# Comparative Study of Graft Healing in 2 Years after “Tension Suspension” Remnant-Preserving and Non-Remnant-Preserving Anatomical Reconstruction for Sherman Type II Anterior Cruciate Ligament Injury

**DOI:** 10.3390/jpm13030477

**Published:** 2023-03-06

**Authors:** Yijia Sun, Zirong Huang, Pingquan Zhang, Huanyu Xie, Chen Wang, Zhenhan Deng, Kang Chen, Weimin Zhu

**Affiliations:** 1Clinical College of the Second Shenzhen Hospital, Anhui Medical University, Shenzhen 518025, China; 2Department of Sports Medicine, Shenzhen Second People’s Hospital, Shenzhen 518025, China; 3Clinical Medical College, Guangzhou Medical University, Guangzhou 511436, China

**Keywords:** anterior cruciate ligament, tension suspension, remnant-preserving reconstruction, graft healing

## Abstract

Purpose: To evaluate the degree of graft healing after “tension suspension” reconstruction of “Sherman II” anterior cruciate ligament injuries versus non-remnant preserving anatomical reconstruction and to compare the clinical outcomes of the two procedures. Method: The clinical data of 64 patients were retrospectively included. There were 31 cases in the “tension suspension” remnant-preserving reconstruction group and 33 cases in the non-remnant-preserving anatomical reconstruction group. The International Knee Documentation Committee (IKDC) score, the Tegner score, and the Lysholm activity score were assessed preoperatively and at 6 months, 1 year, and 2 years postoperatively, respectively. The signal/noise quotient (SNQ) of the grafts was measured at 6 months, 1 year, and 2 years after surgery to quantitatively evaluate the maturity of the grafts after ACL reconstruction; the fractional anisotropy (FA) and apparent diffusion coefficient (ADC) of the reconstructed ACL region of interest (ROI) were measured using DTI. Result: A total of 64 patients were included in the study. The mean SNQ values of the grafts in the 6 months, 1 year, and 2 years postoperative remnant-preserving reconstruction (RP) groups were lower than those in the non-remnant-preserving (NRP) reconstruction group, with a statistically significant difference (*p* < 0.05). At each postoperative follow-up, the SNQ values of the tibial and femoral sides of the RP group were lower than those of the NRP group; the SNQ values of the femoral side of the grafts in both groups were higher than those of the tibial side, and the differences were statistically significant (*p* < 0.05). At 6 months, 1 year, and 2 years postoperatively, the FA and ADC values of the grafts were lower in the RP group than in the NRP group, and the differences were statistically significant (*p* < 0.05); the IKDC score and Lysholm score of the RP group were higher than the NRP group, which was statistically significant (*p* < 0.05). Conclusion: For Sherman II ACL injury, the graft healing including ligamentization and revascularization at 2 years after the “tension suspension” remnant-preserving reconstruction was better than that of non-remnant-preserving anatomic reconstruction.

## 1. Introduction

The anterior cruciate ligament (ACL) is an important stabilizing structure of the knee joint, and injury can lead to knee instability, secondary to meniscal and cartilage damage, which in turn causes accelerated degeneration of the knee [1]. Although ACL reconstruction is the current “gold standard” for the treatment of ACL injuries, there are still some problems that remain unsolved, even though satisfactory results have been achieved. Recent research has shown that the incidence of osteoarthritis is higher and the age of onset is significantly earlier in patients with ACL reconstruction compared to the general population [2]. Most scholars believe it is related to the fact that the stability of the knee joint has not been fully restored to normal after ACL reconstruction and there are still abnormalities in the biomechanics of the lower limb [3,4].

Recently, the preservation of the remnant of the ACL has become a hot spot in the treatment of anterior cruciate injuries, and there are different opinions on whether to preserve the remnant of the ACL. Some scholars believe that it contains a large number of residual blood vessels and various tissue cells, and can provide the required nutritional support and a favorable growth environment for the tendon-bone interface after ACL reconstruction, and the residual proprioceptors in it have a good effect on the proprioceptive recovery and functional recovery of the knee joint [5,6,7]. However, some authors believe that the residual tissue can produce inflammatory factors that can adversely affect the prognosis of the surgery and have an influence on the intraoperative positioning of the graft, which can be negative for the healing of the tendon-bone after surgery [8,9,10]. Although the studies reported have observed a number of complications associated with remnant-preserving reconstruction techniques, our team believes that strict surgical indications will prevent adverse outcomes. We therefore strictly selected remnants of Sherman type II or higher to ensure that sufficient remnants were available for tension suspension reconstruction surgery.

However, there is still a lack of uniform, quantitative, non-invasive methods for assessing the healing of grafts after ACL reconstruction [11]. Magnetic resonance imaging (MRI) is an important examination to analyze the anatomic position, ligamentization, revascularization, mechanical properties, and prognosis of the graft. ACL reconstruction with remnant preservation is a current research hotspot in sports medicine, and some scholars believe that preserving the remnant can accelerate graft revascularization and promote knee proprioceptive recovery [12]. There are many reports of clinical follow-up studies of remnant-preserving reconstruction techniques, but few studies on the assessment of healing of remnant-preserving reconstructed grafts on MRI! Therefore, this study reviewed and analyzed the cases of Sherman II anterior cruciate ligament injury, and compared the healing of the graft on MRI 2 years after the “tension suspension” method of remnant-preserving reconstruction and non-remnant-preserving anatomical reconstruction, in order to better guide the clinical treatment and postoperative rehabilitation.

## 2. Materials and Methods

This was a retrospective comparative study conducted at a single institution. We reviewed patients who underwent ACL reconstructive surgery at the Department of Sports Medicine, The First Affiliated Hospital of Shenzhen University from May 2017 to March 2019. Inclusion criteria were (1) intraoperative arthroscopic finding of an ACL tear of type Sherman II with synovial coverage and an intact tibial attachment; (2) complete preoperative examination data; (3) access to more than 2 years of postoperative follow-up; and (4) completion of surgery by the same senior surgeon. Exclusion criteria: (1) history of knee surgery; (2) combined posterior cruciate ligament injury, lateral collateral ligament injury, or medial collateral ligament injury of grade III or higher; (3) severe cartilage injury (Outerbridge classification grade 3or 4); (4) lower limbs malalignment; (5) joint hypermobility syndrome.

Based on these criteria, a total of 64 patients with “Sherman II” type ACL injury, 39 males and 25 females, aged 33.5 ± 10.4 years, were included. The mean time from injury to surgery was 16 (1 to 32) weeks, and the follow-up time for all patients was 18.5 ± 5.0 months (Table 1). Thirty-one cases in the remnant preservation (RP) reconstruction group, the anatomic insertion of the femur and the posterior position of the tibial anatomical insertion were selected for single-bundle reconstruction; 33 cases in the non-remnant preservation (NRP) anatomical reconstruction group, selected the central position of the femoral and tibial anatomical insertion for single-bundle reconstruction (Figure 1).

## 3. Surgical Technique

Tension, suspension remnant preservation reconstruction: (1) arthroscopic assessment of the remnant was Sherman II type, and the femoral insertion was selected below the lateral intercondylar ridge (resident ridge) and positioned with the resident ridge as the center of the bone tunnel; to protect the tibial insertion of the ACL, the tibial insertion was selected slightly posterior to the ACL remnant, close to the center of the PL bundle. (2) Using a shoulder joint suture grasper, 2 stitches of high-strength suture were placed proximal to the ACL remnant (continuous suture method), and the high-strength suture was pulled through the tibial tunnel with a wire penetrating grasper and threaded into the hole of the Endobutton. (3) Two strands of suture are formed through the Endobutton and folded in half as the tension-reducing suture, which passes through the bone tunnel with the graft, the tension-reducing suture is gradually tightened at the remnant while maintaining tension, and uses the shoulder joint knot pusher to tie and fix the knot at the surface of the Endobutton (Figure 2). Non-remnant preservation anatomical reconstruction: the ACL remnant was excised to fully expose the femoral and tibial insertions, with the femoral insertion chosen at the anatomic center point below the resident ridge and slightly posterior to the lateral bifurcate ridge, and the tibial insertion chosen at the center of the tibial remnant. All patients were fixed with the femoral Endobutton and the tibial peek compression screw (Smith & Nephew Inc.). (Figure 3).

## 4. Rehabilitation Protocol

Postoperatively, patients performed straight leg raising and ankle pump exercises in bed, and started active knee flexion and passive knee extension training with a knee brace after 72 h. Active knee flexion and passive knee extension training continued from 4 to 12 weeks, and the knee flexion angle gradually increased to 90°–120° at 4 weeks, and quadriceps muscle strength exercises were strengthened. Full range of motion of the knee joint is possible at 3–6 months, depending on the patient’s tolerance to continue weight-bearing walking, with a gradual increase in activity and avoiding strenuous exercise; swimming and cycling are possible at 6 months after surgery, jogging resumes at 10 months, and participation in antagonistic sports is possible after 1.5 years [13].

## 5. Clinical Assessment

Assessments included the International Knee Documentation Committee (IKDC) score, Tegner score, and Lysholm activity score. These assessments were performed and recorded before and 6 months, 1 year, and 2 years after surgery to assess knee function and tibial anterior displacement.

## 6. Radiographic Evaluation

### 6.1. Signal/Noise Quatient (SNQ) Measurement

MRI (Siemens, Germany) was performed on the affected knee at 6 months, 1 year, and 2 years postoperatively. Scan conditions: repetition time/echo time 3000/41 ms, field of view 15 cm × 15 cm, matrix 240 × 320, layer thickness 3.0 mm. The images were imported into RadiAnt DICOM viewer 5.0 software (Medixant Inc., Poland), and the sagittal suppressed fat imaging intermediate level images were taken, and the signal intensity was measured in the selected areas of the femoral insertion of the graft, the tibial insertion of the graft, the quadriceps, and the background (2 cm in front of the patellar ligament). The area of interest is 0.2 cm^2^, and all area of selected sites is the same (Figure 1). The signal intensity at each graft site is quantified and the SNQ value is calculated according to the following formula: (signal intensity at each graft site—quadriceps signal intensity)/background signal intensity [13]. The mean value of the SNQ value at the tibial and femoral insertions was used as the graft SNQ value. Measurements were taken by 2 physicians separately, two times in total, 2 weeks apart, and the mean of the 2 results was taken for analysis.

### 6.2. DTI Measurement

DTI parameters: axial scanning with 32 directional gradient magnetic fields, b-value 400 s/mm^2^, TE90 ms, TR2500 ms, layer thickness 2.9 mm, layer spacing 0 mm, field of view 180 mm × 180 mm, matrix 128 × 128, excitation number 4, scan time 4 min.

Two physicians experienced in diagnostic imaging performed a double-blind reading of the images, independently measured, analyzed and agreed on a uniform result, all measurements are completed at the GE AW 4.6 post-processing workstation. The DTI images were automatically corrected using Functool software (Version 4.6,General Electric Company Healthcare, USA), which then generated pseudo-color images of the sequence.

The DTI images were fused with the transaxial FSE PDWI images to form an anatomical localization map. The region of interest (ROI) was manually selected based on the anatomical localization map, and the ROI was placed along the ACL travel area and avoided other structures, and the fractional anisotropy (FA) and apparent diffusion coefficient (ADC) values were measured. The same area was measured again in each case of ACL injury corresponding to the reference group. All measurements were repeated three times and the average was taken.

### 6.3. Statistical Analysis

Data were collected and analyzed using SPSS 25.0 statistical software (Version 25.0,International Business Machines Corporation, USA). Data were expressed as mean ± standard deviation. Independent samples *t*-test was used to compare continuous variables, including IKDC, Tegner, and Lysholm scores, and differences between the remnant-preserving and non-remnant-preserving groups at the same time point and the same site at SNQ values, ADC values, and FA values. *p* < 0.05 differences being statistically significant. Efficacy analysis was performed using G*Power software (Version 3.1.9.7,University of Düsseldorf, Germany). A two-independent samples *t*-test was used calculation, with the significance level α set at 0.05 and 1-β at 0.8. A review of the literature revealed valid values for each knee function score, with the highest calculated effect size of 2.4 for the Lysholm subjective knee function score [14,15]. The final calculation resulted in a minimum sample size of 32 cases, with 16 cases in each group; when the sample size ratio between the two groups was less than 2, the minimum sample size was 36 cases, with at least 12 cases in each group.

## 7. Result

### 7.1. Clinical Functional Evaluation Results

As shown in Table 2, the results of all functional scores measured at 6 months, 1 year, and 2 years after the remnant preservation reconstruction and non-remnant preservation anatomical reconstruction in this study were significantly improved compared with those before surgery (*p* < 0.01). Compared with 6 months, the objective score at 12 and 24 months also improved significantly (*p* < 0.01), and the functional score of the RP group was better than that of the NRP group, with a statistically significant difference (*p* < 0.01).

### 7.2. Radiographic Results

The mean SNQ values of the grafts were 16.517 ± 6.272, 12.624 ± 5.987, and 9.902 ± 6.201 at 6 months, 1 year, and 2 years postoperatively for the RP group, and 24.407 ± 6.173, 15.721 ± 5.882, and 11.901 ± 6.216 for the NRP group. The differences were statistically significant (t = 1.827, *p* = 0.008). At the same postoperative time point at the same site of the graft, SNQ values were greater in the NRP group than in the RP group, with statistically significant differences (*p* < 0.05), (Table 3). The FA values of the grafts were 0.329 ± 0.041, 0.237 ± 0.032, and 0.161 ± 0.036 at 6 months, 1 year, and 2 years postoperatively for the RP group, and 0.376 ± 0.043, 0.250 ± 0.034, and 0.176 ± 0.029 for the NRP group, respectively, with significant differences between them (*p* < 0.001). The ADC values of the grafts in the RP group were 2.812 ± 0.161 (×10^−3^ mm^2^/s), 2.510 ± 0.143 (×10^−3^ mm^2^/s), and 2.012 ± 0.321 (×10^−3^ mm^2^/s) at 6 months, 1 year, and 2 years postoperatively, and the NRP group were 2.911 ± 0.159 (×10^−3^ mm^2^/s), 2.621 ± 0.138 (×10^−3^ mm^2^/s), and 2.214 ± 0.291 (×10^−3^ mm^2^/s), respectively, with significant differences between them (*p* < 0.005) (Table 4).

## 8. Discussion

The greatest highlight of this study is that it provides a new clinical treatment thinking and demonstrates that remnant-preserving reconstruction has better imaging results and clinical outcomes compared to traditional ACL reconstruction, provided that the integrity of the remnant is ensured through strict screening of indications. This study demonstrates the importance of refined and individualized treatment through the application of more appropriate treatment protocols for different patients.

It was found [12] that some remnants are left in 58% of ACL tears, with the injury occurring mostly on the femoral stop side and the tibial side being relatively intact. The presence of vascular bundles, mesenchymal stem cells, synovial membrane, and neurons in the remnant is beneficial to the recovery of proprioception of the graft ligament, revascularization, and reinnervation of neurons, which has a positive effect on the recovery of the graft ligament shape and function; moreover, the presence of the remnant is beneficial to the accurate positioning and the closure of the internal port of the bone tunnel, which prevents the leakage of joint fluid, reduces the enlargement of the bone tunnel and promotes the healing of the ligament to bone; meanwhile, the wrapping of the remnant around the graft also increases the stability of the joint to a certain extent [16,17,18].

MRI has become the first choice for assessing tendon graft healing and the ligamentization process after ACL reconstruction because of its significant advantages such as non-invasive, convenient, and digital analysis [19,20]. During the healing process, the internal revascularization and water content of the graft change, and its MRI signal intensity varies over time. Therefore, the signal/noise quotient (SNQ value) based on MRI signal intensity can be considered as a quantitative assessment of the degree of healing! The degree of graft healing was assessed by measuring SNQ values at different sites of the ligament, and lower SNQ values demonstrate a better degree of healing in ACL grafts because internal blood flow is reduced after ligamentization of tendon grafts [21]. Wang [22] et al. measured the signal intensity of the proximal, middle, and distal ligament grafts by MRI and calculated the SNQ values to assess the ligamentization, respectively, and showed that the SNQ values of the proximal, middle, and distal ligament grafts gradually decreased with time, and the SNQ values of the middle ligament showed a significant decrease at 2 years postoperatively and the SNQ values of the proximal and distal ligaments at 4 years postoperatively. In this study, MRI imaging studies of the tension suspension RP group and the NRP group at 6 months, 1 year, and 2 years after reconstruction revealed that both the SNQ values of the proximal, middle, and distal ligament grafts showed a gradual decrease with time, and the SNQ values of the tension suspension RP group were lower than the NRP group, suggesting a better ligamentization process. The reason may be that the RP reconstruction technique promotes blood supply further coverage of the remnant to the graft, while the NRP ligament needs to rely on synovial coverage to provide blood supply, so the ligamentization of the NR reconstruction is significantly faster than NRP reconstruction. In the RP reconstruction technique, the femoral side of the remnant is often not completely covered, resulting in poorer blood coverage at the femoral side than at the tibial side, and this part requires synovial coverage to provide blood supply, but the synovial coverage takes longer, resulting in longer ligamentization time at the femoral side, so the SNQ values at the femoral side takes a relatively long time to stabilize. Systematic review studies have shown [23,24,25] that the graft SNQ values peaks at 6 months after ACL reconstruction and gradually decreases to normal ligament level. In this study, the highest SNQ values was found at 6 months postoperatively in both groups, which is a consistent view. Therefore, 5-6 months postoperatively is the best time for early assessment of the degree of ligamentization of grafts [26,27,28].

The DTI technique is based on the principle of anisotropy in the diffusive motion of water molecules and quantitatively evaluates the pathological changes in tissue microstructure by applying unique parameters such as fraction anisotropy (FA) values and apparent diffusion coefficient (ADC) values, while generating diffusion tensor tractography (DTT) images to accurately evaluate the fibrous tissue structure [29]. The FA value indicates the proportion of the anisotropic component of water molecules occupying the whole diffusion tensor, reflecting the degree of spatial displacement of water molecules in the direction of fiber bundles in tissues; the larger the FA value, the higher the degree of anisotropy; the ADC value is used to measure the state of diffusion movement of water molecules in the human tissue environment, reflecting the intensity of displacement of water molecules in the direction of diffusion-sensitive gradients, the larger the ADC value, the stronger the diffusive movement of water molecules in the tissue. In this study, the ADC and FA values at 6 months, 1 year, and 2 years postoperatively were higher in the NRP group than in the RP group, and the ADC and FA values in both groups gradually decreased over time. It is possible that the FA values are related to the direction of fiber bundles in the tissue and that the lower FA values indicate less fiber anisotropy and better remodeling of the ligament, the ADC value decreases gradually as the new vessels grow into the graft and the graft tends to mature. Chen [30] et al. used DTI images to perform fiber tracer imaging of the ACL and measured the FA and ADC values of the femoral side, middle segment, and tibial side of the normal ACL. The results showed that the ADC values were lower in the normal group than in the ACL injury; the FA values were higher in the normal group than in the injury group. Pieter [29] et al. also used DTI as a visual and quantitative parameters to assess graft healing after ACL reconstruction. In this study, by measuring FA and ADC values at 6 months, 1 year, and 2 years time periods in both groups, it was found that both showed a gradual decrease with time, and the values measured at 2 years postoperatively in the tension suspension RP group were closer to normal ACL with better shaping. The research by Yang [31] et al. found significantly higher FA values in patients 10 years after ACLR reconstruction than in other patients with a shorter period after reconstruction, so we need to further follow up this study population to obtain long-term observations. At the postoperative follow-up assessment of knee function, it was found that the IKDC score and Lysholm score of the RP group were higher than those of the NRP group, which was statistically significant (*p* < 0.05), and that the RP group could recover better knee function postoperatively compared to the NRP group.

Combined with the imaging and functional assessment, we believe that stump-preserving reconstruction of the ACL in this study is the recommended surgical technique. In the previous technique, the remnant was directly sutured together with the graft, and the remnant was not fixed with a certain tension, resulting in a poor bonding of the remnant with the graft or even free, resulting in a “Cyclops lesions” leading to impaction; if the healing between the remnant and the graft is poor, the gradual resorption of the remnant also loses its usefulness. In this study, the above problems can be solved by continuous suturing of the proximal side of the remnant, and then pulling the tension-reducing thread through the Endobutton and into the femoral tunnel together with the graft, and tying a knot on the surface of the Endobutton to maintain the tension of the remnant. Technical points: (1) Without affecting the visual field and bone tunnel positioning, try not to destroy the remnant and the synovial sheath and subpatellar fat pad tissue, so as to maximize the functional role of the remnant and reduce the postoperative intra-articular scar formation. (2) The intraoperative suture tension on the remnant should be moderate to maintain the stability to prevent the “Cyclops lesions”. (3) The tibial tunnel is drilled slightly posterior to the ACL remnant near the center of the PL bundle, with the tibial locator angled at 50° to both protect the remnant and facilitate the graft being located posterior to the majority of the remnant; and when the drill stops just after penetrating the cortical bone, the hole is reamed slightly posteriorly from the anterior edge of the remnant, taking care to preserve as much of the remnant tissue and the surface synovial sheath as possible. (4) The remnant is prepared as a “cuff” and the graft should be positioned posterior to the remnant to avoid impaction. (5) The femoral tunnel is positioned with the knee flexed at 90°, the ACL femoral insertion and the posterior cartilage margin of the posterior femoral epicondyle are fully exposed with the shaver, the highest point of the cartilage margin and the midpoint of the remnant crossed for reference of the location point, then flex the knee at 120° and drill the femoral tunnel. (6) A knot pusher was used to maintain tension knot fixation at the surface of the Endobutton to maintain a tight fit between the remnant and the graft tendon, which facilitates graft crawl replacement and tendon bone healing and increases joint stability.

## 9. Limitation

The small sample size of this study limits the expansion and expansion of data in this study to some extent, and the next step needs to increase the sample size of the study in order to find more reliable conclusions. Lack of arthroscopic secondary examination and histological analysis to accurately assess graft healing. This study is a retrospective case-control study and cases may be subject to selection bias. The follow-up period is relatively short, and further long-term follow-up is needed to understand the recovery of kinematics and biomechanics, proprioception. Relatively few studies have been reported, and the settings of SNQ and DTI scan parameters are not completely unified because of the different MR devices.

## 10. Conclusions

For the “Sherman II” ACL injury, the “tension suspension” remnant-preserving reconstruction can well restore the stability and function of the knee joint, and the 2 year follow-up MRI performance is better than that of non-remnant preserving anatomical reconstruction. The next step will be to continue the follow-up to clarify the long-term MRI performance to better guide the clinical treatment.

## Figures and Tables

**Figure 1 jpm-13-00477-f001:**
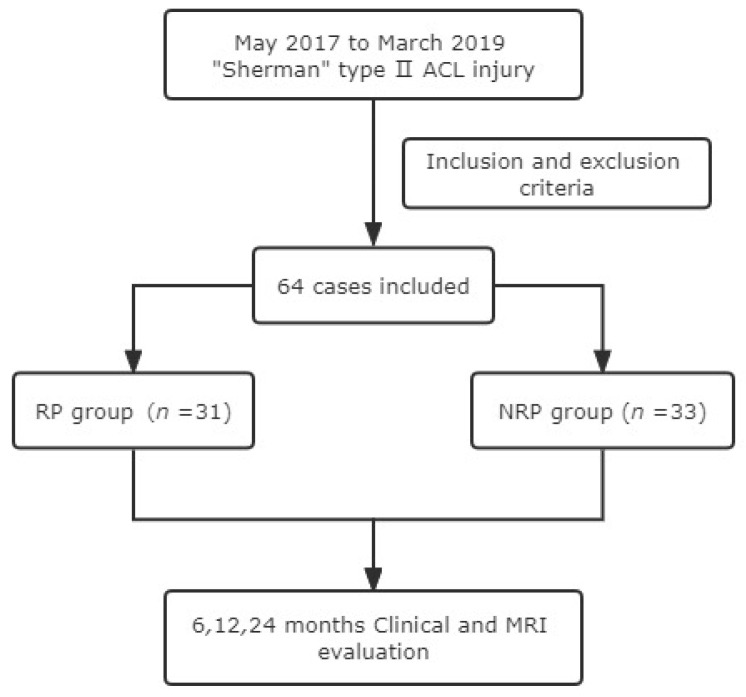
Patient flow diagram.

**Figure 2 jpm-13-00477-f002:**
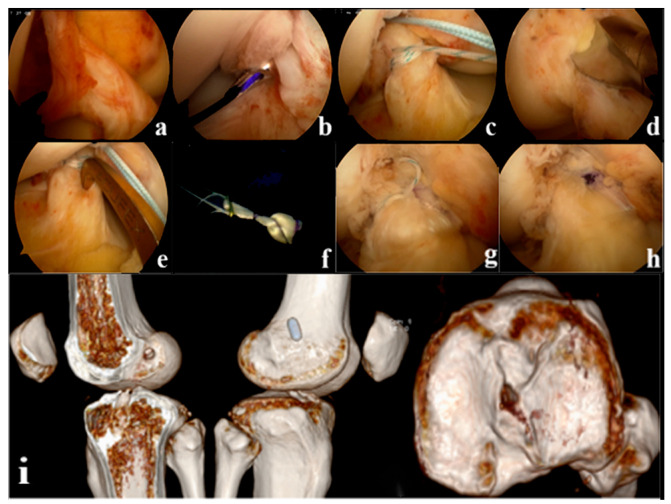
(**a**): Arthroscopic appearance of the anterior cruciate ligament (ACL) remnant (**b**): Suture at the proximal side of the remnant where the synovium is intact (**c**): Continuous suture closure of the remnant (**d**): Positioning the femoral tunnel (**e**): Positioning the tibial tunnel (**f**): External threading of tension-reducing wires (**g**): The remnant is wrapped in a sutured sleeve with the graft and fixed with tension-reducing wire to maintain tension (**h**): Final arthroscopic appearance of the tension suspension remnant-preserving reconstruction (**i**): Postoperative Computed Tomography (CT) showed that the ACL reconstructed bone tunnel was accurately positioned and the tibial bone tunnel was posteriorly positioned close to the center of the PL bundle.

**Figure 3 jpm-13-00477-f003:**
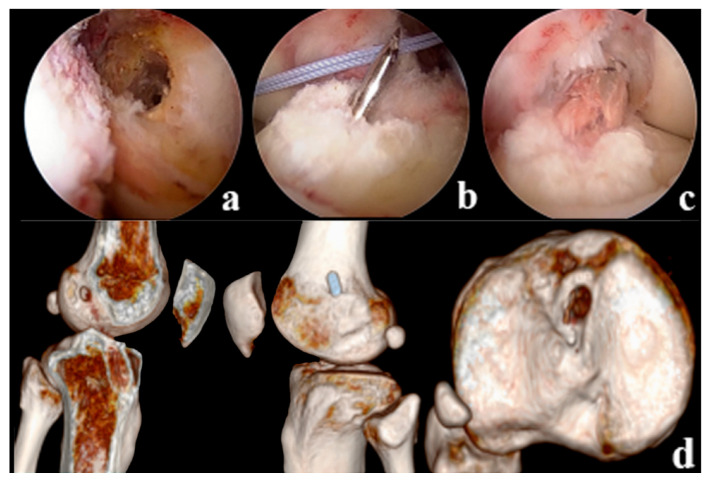
(**a**): Positioning of the anatomically reconstructed femoral tunnel in the NRP group (**b**): Tibial bone tunnel drilling (**c**): ACL Reconstruction (**d**): Postoperative CT showed reconstruction of the anatomical position of the femoral tunnel, with the tibial bone tunnel located in the center of the remnant.

**Table 1 jpm-13-00477-t001:** Characteristics of patients.

	RP Group (*n* = 31)	NRP Group (*n* = 33)	T Value	*p* Value
Age	27.71 ± 7.24	27.26 ± 7.05	−0.648	n.s
Gender, Male/Female	19/12	20/13		n.s
BMI, kg/m^2^	23.61 ± 1.22	23.46 ± 1.17	−0.500	n.s
Injured side, left/right	11/20	14/19		n.s

Note: Statistical differences were found within and between groups (*p* < 0.05).

**Table 2 jpm-13-00477-t002:** Clinical functional evaluation results.

	RP Group (*n* = 31)	NRP Group (*n* = 33)	*t*	*p*
$\mathrm{IKDC}\;(\bar{X}$ ± s)				
Preoperation	49.3 ± 8.4	50.1 ± 7.8	1.568	0.122
6 months	68.7 ± 7.9	66.8 ± 6.4	4.067	0.000 *
12 months	84.1 ± 6.3	82.5 ± 7.1	4.908	0.000 *
24 months	88.6 ± 4.8	86.5 ± 5.1	1.961	0.05 *
$\mathrm{Lysholm}\;(\bar{X}$ ± s)				
Preoperation	51.2 ± 9.1	50.4 ± 10.1	−1.377	0.174
6 months	73.2 ± 6.3	72.4 ± 6.0	2.701	0.009 *
12 months	83.8 ± 5.4	81.4 ± 5.3	5.376	0.000 *
24 months	89.8 ± 4.7	87.6 ± 4.2	1.506	0.03 *
$\mathrm{Tegner}\;(\bar{X}$ ± s)				
Preoperation	2.2 ± 0.8	2.1 ± 0.9	−0.394	0.693
6 months	3.6 ± 0.6	3.5 ± 0.7	−0.699	0.485
12 months	5.3 ± 1.1	5.2 ± 0.8	−1.753	0.080
24 months	6.0 ± 0.7	5.8 ± 0.9	−0.254	0.800

Values are presented as n or mean ± SD. Asterisk value indicates statistical significance (* *p* < 0.05).

**Table 3 jpm-13-00477-t003:** SNQ values of grafts of two groups at different sites and time.

	RP Group			NRP Group			*p* Value		
	6 Months	12 Months	24 Months	6 Months	12 Months	24 Months	6 Months	12 Months	24 Months
Femoral side	21.86 ± 7.51	17.32 ± 6.69	12.16 ± 7.08	29.45 ± 11.47	21.04 ± 7.96	17.31 ± 7.87	*p* < 0.05 *		
Middle segment	11.98 ± 5.51	9.84 ± 6.01	9.32 ± 6.48	21.42 ± 9.31	18.69 ± 7.22	13.45 ± 6.92			
Tibial side	10.02 ± 5.98	9.21 ± 5.23	9.95 ± 6.11	19.76 ± 8.56	13.18 ± 6.54	11.82 ± 6.96			

Values are presented as n or mean ± SD. Asterisk value indicates statistical significance (* *p* < 0.05).

**Table 4 jpm-13-00477-t004:** Mean FA value and ADC value of the two groups at different time.

	RP Group			NRP Group			*p* Value		
	6 Months	12 Months	24 Months	6 Months	12 Months	24 Months	6 Months	12 Months	24 Months
FA value	0.329 ± 0.041	0.237 ± 0.032	0.161 ± 0.036	0.376 ± 0.043	0.250 ± 0.043	0.176 ± 0.029	*p* < 0.05 *		
ADC value	2.812 ± 0.161	2.510 ± 0.143	2.012 ± 0.321	2.911 ± 0.159	2.621 ± 0.138	2.214 ± 0.291			

Values are presented as n or mean ± SD. Asterisk value indicates statistical significance (* *p* < 0.05).

## Data Availability

The datasets used and analyzed during the current study available from the corresponding author on reasonable request. We simply extracted data and did not involve the private information of patients.

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
