# Peer review of "Comparative Study of Graft Healing in 2 Years after “Tension Suspension” Remnant-Preserving and Non-Remnant-Preserving Anatomical Reconstruction for Sherman Type II Anterior Cruciate Ligament Injury"

_jpm, 2023, doi:10.3390/jpm13030477_

Round 1

Reviewer 1 Report

Authors have done a fairly large amount of work on a comparative analysis of anterior cruciate ligament anatomical reconstruction with remnant and non-remnant preservation. This is an interesting topic, but in order to publish research materials as an article in the Q2 journal, the manuscript needs to be seriously revised. Some of changes that need to be made from my point of view are listed below.

1. To design References, you must to use “Instructions for Authors” (https://www.mdpi.com/journal/jpm/instructions).

2. The first time MRI is mentioned, use: Magnetic resonance imaging (MRI).

3. The abstract should be a total of about 200 words maximum (see “Instructions for Authors”).

4. It is necessary to more clearly formulate the purpose of the study. Remove from the paragraph “Purpose” in the Abstract: “MRI was used…” This phrase should be moved to the “Materials and Methods” section.

5. The introduction needs to be expanded. At the same time, it is necessary to substantiate the current state of research on the topic of the article, point out unresolved problems and formulate the purpose of the study in a reasonable manner, with reference to published literary sources. It is desirable to review the problem using more literature published during the last 5 years. Apparently, authors did not analyze the state of the problem in sufficient detail, because many recent reviews were not included in the introduction (see, for example: Xie Huanyu, et al. Effects of remnant preservation in anterior cruciate ligament reconstruction: A systematic review and meta-analysis. Frontiers in Surgery 2022, 9. https BioMed Research International 2019, Article ID 1652901 https://doi.org/10.1155/2019/1652901 Rothrauff, B.B. et al Anterior cruciate ligament reconstruction with remnant preservation: current concepts J ISAKOS 2020, 5, 128-133 https://doi.org/10.1136 /jisakos-2019-000321 and many more.

6. Sections “Material and Methods” and “Results” are presented adequately.

7. In the “Discussion” section, it is necessary to focus on results obtained in this study. The novelty of obtained results should be especially emphasized, how they differ from results obtained earlier by other researchers. Taking into account the subject of the Journal of Personalized Medicine, it should be noted personalized features of the discussed surgical technology. An overview of the problem state of the art should be moved to the “Introduction” section.

Author Response

We feel great thanks for your professional review work on our paper. As you are concerned, several problems need to be addressed. According to your nice suggestions, we have made extensive corrections to our previous manuscript, and the revisions are highlighted in red, the detailed corrections are listed below.

  1. We have revised the references in accordance with the "Instructions for Authors" to ensure that all references are in accordance with the rules.
  2. We have added a note on the first use of MRI.
  3. We have extensively streamlined the abstract section to bring it in line with the "Instructions for Authors".
  4. We have streamlined the descriptions in the Purpose section and detailed the MRI-related descriptions in the Discussion section.
  5. We have extensively revised and extended the Introduction section and cited more recent literature for clarification as you suggested.
  6. The "Materials and methods" and "Results" sections remain unchanged.
  7. We have revised the Discussion section to emphasize the main findings of the study, whose innovative and individualized treatment concept is more in line with the theme of this journal. The innovative points and individualization of the surgical technique are also explained in detail.

Reviewer 2 Report

This study entitled “Comparative study of graft healing in 2 years after "tension suspension" remnant-preserving and non-remnant-preserving anatomical reconstruction for Sherman type II anterior cruciate ligament injury” seems to have been generally well executed and written. Furthermore, I believe that his paper will be of great interest to the readers. Finally, I have only a few minor suggestions to improve the quality of the paper.

Introduction

Throughout the whole Introduction section you have only two references. Please add more references.

Materials and Methods

Statistical analysis

Although, the design of the study is retrospective the sample size should be calculated.

Discussion

Begin this section with the main findings of your study.

Limitation

Please state the Limitation section before the Conclusion section.

Author Response

We feel great thanks for your professional review work on our paper. As you are concerned, several problems need to be addressed. According to your nice suggestions, we have made extensive corrections to our previous manuscript, and the revisions are highlighted in red, the detailed corrections are listed below.

1.Introduction

We have made additions and added sufficient references to illustrate the current situation.

2.Statistical analysis

We have streamlined the description of the statistical analysis and added a sample size calculation.

3.Discussion

We have revised the discussion section to emphasize the main findings of the study, whose innovative and individualized treatment concept is more in line with the theme of this journal.

4.Limitation

We have adjusted the limitations section to precede the conclusions section.

Round 2

Reviewer 1 Report

Despite the fact that not all of previously indicated shortcomings have been eliminated, the article can be recommended for publication in this variant. Authors have the right to present their own point of view.